# Improvement of Stable Restorer Lines for Blast Resistance through Functional Marker in Rice (*Oryza sativa* L.)

**DOI:** 10.3390/genes11111266

**Published:** 2020-10-27

**Authors:** Jegadeesan Ramalingam, Savitha Palanisamy, Ganesh Alagarasan, Vellaichamy Gandhimeyyan Renganathan, Ayyasamy Ramanathan, Ramasamy Saraswathi

**Affiliations:** 1Department of Biotechnology, Agricultural College and Research Institute, Tamil Nadu Agricultural University, Madurai 625104, India; vgrenga@gmail.com; 2Centre for Plant Molecular Biology and Biotechnology, Tamil Nadu Agricultural University, Coimbatore 641003, India; saviagri@gmail.com (S.P.); alagarasan.ganesh@hotmail.com (G.A.); 3Department of Rice, Centre for Plant Breeding and Genetics, Tamil Nadu Agricultural University, Coimbatore 641003, India; nathanram@rediffmail.com (A.R.); sarasrice2004@yahoo.co.in (R.S.)

**Keywords:** rice, blast, fertility restoration, functional marker, marker-assisted back-cross breeding

## Abstract

Two popular stable restorer lines, CB 87 R and CB 174 R, were improved for blast resistance through marker-assisted back-cross breeding (MABB). The hybrid rice development program in South India extensively depends on these two restorer lines. However, these restorer lines are highly susceptible to blast disease. To improve the restorer lines for resistance against blasts, we introgressed the broad-spectrum dominant gene *Pi54* into these elite restorer lines through two independent crosses. Foreground selection for *Pi54* was done by using gene-specific functional marker, *Pi54 MAS,* at each back-cross generation. Back-crossing was continued until BC_3_ and background analysis with seventy polymorphic SSRs covering all the twelve chromosomes to recover the maximum recurrent parent genome was done. At BC_3_F_2_, closely linked gene-specific/SSR markers, DRRM-*RF3*-10, *DRCG-RF4*-8, and RM 6100, were used for the identification of fertility restoration genes, *Rf3* and *Rf4,* along with target gene (*Pi54*), respectively, in the segregating population. Subsequently, at BC_3_F_3_, plants, homozygous for the *Pi54* and fertility restorer genes (*Rf3* and *Rf4*), were evaluated for blast disease resistance under uniform blast nursery (UBN) and pollen fertility status. Stringent phenotypic selection resulted in the identification of nine near-isogenic lines in CB 87 R × B 95 and thirteen in CB 174 R × B 95 as the promising restorer lines possessing blast disease resistance along with restoration ability. The improved lines also showed significant improvement in agronomic traits compared to the recurrent parents. The improved restorer lines developed through the present study are now being utilized in our hybrid development program.

## 1. Introduction

The major goal of rice production is to fulfill the demand for the growing population and improve food security. Among various genetic approaches available today, hybrid rice technology is the most promising and accepted strategy for improving the rice productivity [1]. However, the hybrid rice production system suffers from various biotic stresses including fungal and bacterial diseases [1,2]. Among which, fungal blast (*Magnaporthe oryzae*), caused by an ascomycete fungus, is one of the major biotic diseases of rice contributing to yield loss up to 10–30% globally [3,4,5,6,7]. In India the disease, under severe and favorable conditions, can cause yield losses ranging from 74 to 100% [8,9,10]. The survivability and high multiplication ability of fungal blasts even in harsh environmental conditions further limit the control of the disease [11]. Deployment of host plant resistance against biotic stress is considered to be a viable strategy for managing the disease in hybrid rice production [12]. However, there is a paucity of high-yielding, disease resistant hybrids/varieties available currently. Up to now, more than 100 genes conferring resistance to blasts have been identified on the rice chromosomes 6, 11, and 12; however, very few of them were introgressed into popular rice varieties [12]. The high variable nature of the pathogen *M. oryzae* leads to frequent emergence of new virulent races resulting in loss of resistance within 3–5 years of cultivation. Currently, *Pi54*, a major dominant resistant (R) gene reported to confer broad spectrum resistance against geographically diverse strains of *M. oryzae* [2,13,14]. The *Pi54* gene located on chromosome 11 has a unique zinc finger domain, besides LRR (Leucine-rich repeat) domain [15,16]. The gene triggers up-regulation of many defense response genes and transcription factors during disease reaction [17]. At the same time, this cytoplasm-localized NBS-LRR domain type (R) gene does not express constitutively in the plant tissues and is expressed only during the recognition of a pathogen-associated molecular pattern (PAMP) from the pathogen [18]. Considering these, *Pi54* was selected as the target resistance gene for introgression into the restorer lines, CB 87 R and CB 174 R.

The general commercial rice hybrid seed production in India follows a three-line approach involving a male sterile (CMS) line, a maintainer line, and a restorer line. Two popular rice hybrids, CO RH 3 and CO RH 4, released by the Tamil Nadu Agricultural University for cultivation, were reported to be susceptible against blasts and bacterial blight (BB) diseases. Recently, the male sterile lines CO 2A, CO 23 A, and CO 24 A (involved in the development of rice hybrids) were introgressed bacterial blight-resistance genes, *xa5, xa13,* and *Xa21,* through functional marker-assisted back-cross breeding (MABB) [19]. However, the restorer lines utilized for hybrid development remain susceptible to both blasts and bacterial blight. Therefore, in the present study, we report the improvement of restorer lines to develop hybrids resistant to blasts. The donor parent, B 95 (a line developed from B 95-1 x Tetep), carrying the *Pi54* gene, tested through All-India Co-ordinated Plant Pathology trials across India has demonstrated broad-spectrum disease resistance against blasts (AICRIP, 2002–2014). Based on this confirmation, the MAS (Marker-assisted selection) program was formulated to introgress the blast resistance gene (*Pi54*) while maintaining fertility restoration (*Rf3* and *Rf4*) genes into popular restorer lines of CB 87 R and CB 174 R in order to develop agronomically superior hybrid rice genotypes for both fungal blasts and bacterial blight resistance.

## 2. Materials and Methods

### 2.1. Plant Materials

Two stable restorer lines, CB 87 R and CB 174 R, which are predominantly employed in the hybrid rice program in South India, were used as recurrent parents. Utilizing these lines, two rice hybrids, CO RH 3 (CO 2A/CB 87 R) and CO RH 4 (CO 23 A/CB 174 R), were recently released from this center for farmers’ cultivation. These hybrids are mostly preferred by the farmers of South India, because of their high seed setting and high yield (6600 and 7348 kg/ha, respectively), but they are highly susceptible to blast infection. The line B 95 (B 95-1 × Tetep) was used as the donor parent for *Pi54* in two independent back-cross breeding steps, and two restorer lines, CB 87 R and CB 174 R, as the recurrent parents. In addition, Tetep as positive check and IR 24 as negative check were used in the screening program. The schematic steps for the introgression of blast resistance genes into restorer lines are presented in Figure 1.

### 2.2. Marker-Assisted Selection for Blasts and Fertility Restoration Genes

Marker-assisted back-cross breeding program was adopted to transfer *Pi54* into two stable restorer lines, CB 87 R and CB 174 R. The fungal blast susceptible restorer lines, CB 87 R and CB 174 R, were crossed with the donor resistant line, B 95. Back-crossing was carried out up to BC_3_ generation. The PCR-based functional marker, *Pi54* MAS [20], was used to identify the heterozygous nature of hybrids at BC_1_F_1_ and subsequent back-cross generations (BC_2_ and BC_3_) (Table 1). One hundred and fifty-six SSR (simple sequence repeat) markers (Appendix A) were used to recover the recurrent parent background in each back-cross generation. A single BC_3_F_1_ plant with high recurrent genome in both the crosses was self-pollinated to produce BC_3_F_2_. The segregating population was screened with *Pi54* MAS to identify homozygous plants. In addition, to verify fertility restoration genes, *Rf3* and *Rf4*, linked/gene-specific markers were used (Table 1).

### 2.3. PCR Amplification

DNAs were extracted from the plants belonging to the parents and back-cross generations employing the simplified mini-scale method [23]. PCR reactions (15 µL) containing 3 µL of genomic DNA, 1× assay buffer, 200 µL of dNTPs, 2 µM MgCl_2_, 0.2 µM each primer (forward and reverse), and 1 unit of *Taq* polymerase (Bangalore Genei) were amplified. The temperature cycle was programmed as 95 °C for 2 min, 94 °C for 45 s, 52 °C for 1 min, 72 °C for 1.30 min for 35 cycles, and an additional temperature of 72 °C for 10 min for extension and 4 °C for cooling. The amplified products were electrophoresed in a 3% agarose gel and stained with ethidium bromide. Clearly resolved, unambiguous bands were scored visually for their presence with each primer.

Closely linked fertility restoration markers, i.e., RM 6100 and DRCG-*RF4-*8 for *Rf4* locus and DRRM-*RF3*-10 for *Rf3* locus, were used to identify the plants with restorer alleles in BC_3_F_2_ generation.

### 2.4. Screening for Blast Resistance, Pollen Fertility, and Evaluation of Agronomic Characters in Improved Lines

The back-cross-derived lines (BC_3_F_3_) along with parents and IR 24 (susceptible check) and Tetep (resistant check) were screened for blast resistance under a uniform blast nursery (UBN) using a virulent local isolate IS (KUL)-6 of *M. oryzae* [24]. For UBN screening, each BC_3_F_3_ line was sown in a single row 50 long and 10 cm apart, and after every 20 test entries and border rows of all the sides of bed the susceptible check IR 24 was planted. This helps to spread the inoculum. Relative humidity is maintained with water sprinklers. The beds were covered during the night to maintain a high humidity until disease development. Additionally, to create severe incidence across the bed, seven-day-old fungal culture maintained as described by Srinivas Prasad et al. [25] was used for artificial inoculation. The adjusted spore concentration of 10^5^ spores/mL [2] was sprayed over the entire UBN plot until the entire plant surface of each individual plant become wet. On the fifteenth day after inoculation, the lines were scored for their resistance reaction as per IRRI (International Rice Research Institute) standard evaluation system using the 0–9 disease severity scale (IRRI 1996).

Pollen fertility was estimated from BC_3_F_3_ lines by using 1% potassium iodide solution (Appendix A). Plants with round and deeply stained pollen grains were classified as fertile. To evaluate the important agronomic traits, twenty-five-day-old seedlings of improved lines along with parents and checks were raised in experimental plot at the Agricultural College and Research Institute, Tamil Nadu Agricultural University, Madurai, India. The experimental plot was arranged in a randomized block design with three replications following standard agronomic practices [26,27]. Each improved line was raised in three rows, each row containing 20 plants. Ten plants in each replication were monitored for the observation of major yield contributing traits. Agronomic traits including days for 50% flowering, plant height (cm), number of productive tillers per plant, panicle length (cm), number of filled grains, 100-grain weight (g), and grain yield per plant (g) were recorded.

### 2.5. Statistical Analysis

The segregation pattern of *Pi54* gene was calculated using the chi-square test (χ^2^) with the formulae χ^2^ = (O−E)^2^/E, where O represents the observed frequency and E represents the expected frequency. The SAS (Statistical Analysis System) package (SAS Institute Inc., Cary, NC, USA) was used for the analysis of the significance of variation between parents and improved restorer lines. The agglomerative clustering among the selected pyramided lines and parents was calculated based on seven quantitative traits using “R” software (R, 2013).

## 3. Results

The F_1_ plants in two crosses were confirmed for their heterozygous nature using the functional marker, *Pi54* MAS. Heterozygous plants were back-crossed to respective recurrent parents (CB 87 R and CB 174 R) to develop BC_1_F_1_. In BC_1_F_1_ generation, a total of 21/183 (Imp. CB 87 R) and 17/194 (Imp. CB 174 R) plants showed heterozygosity for the marker, *Pi54 MAS,* which was linked to the blast resistance gene. The positive BC_1_F_1_ plants were subjected to background selection using 70 polymorphic markers (https://archive.gramene.org/markers/microsat/ssr.html) to identify the plants with maximum recurrent parent genome (R_PG_R). The heterozygous plants with a maximum parental genome of 73.6% (CB 87 R) and 74.9% (CB 174 R) were selected and back-crossed to generate BC_2_F_1_ plants. Back-crosses were continued up to BC_3_ to recover the maximum recurrent genome. Positive BC_3_F_1_ plants with the maximum recurrent genome of 91.1% in CB 87 R and 90.3% were self-pollinated to produce BC_3_F_2_ plants. A total of 41 homozygous plants in CB 87 R and 46 in CB 174 R for the *Pi54* MAS marker were screened for fertility restoring genes (*Rf3* and *Rf4*) using gene specific/linked markers; DRRM-*RF3*-5 for *Rf3* and DRCG-RF4-8 & RM 6100 for *Rf4*. Nine (CB 87 R) and 13 (CB 174 R) plants showing the co-segregating banding pattern in BC_3_F_3_ (Figure 2) were subjected to the pollen fertility study and agronomic traits. The number of plants scored for both blasts and fertility restoration using gene-specific markers in different back-crosses generation is provided in Appendix A. The segregation pattern of blast resistance in BC_3_F_2_ generation plants was tested using χ^2^ statistics. In the CB 87 R × B 95 cross, the observed segregating frequency was 41:106:63, while in the other cross, CB 174 R × B 95, it was about 46:127:87. Collectively, in both the crosses, the χ^2^ value was non-significant and hence the observed genotypic ratio was nearly equal to the expected genotypic ratio (Table 2).

### 3.1. Assessment of Blast Resistance in the Pi54 Gene-Introgressed Lines

Twenty-two three-gene homozygous back-cross-derived lines were included for blast screening in the cross combination, along with recurrent parents, donor, and positive (Tetep) and negative check (IR 24). The disease susceptibility was evaluated for blast pathogen under controlled artificial greenhouse environment. The donor parent, B 95, and positive check, Tetep, harboring the *Pi54* gene, showed a high level of resistance to rice blasts (Figure 3) and scored as “1”, whereas, CB 87 R and CB174 R showed susceptibility to rice blasts and were scored as “9”. The twenty-two advanced three-gene (*Pi54 + Rf3 + Rf4*) homozygous-introgressed lines viz. Imp. CB 87 R (9 lines) and Imp. CB 174 R (13 lines) were selected from both of the parental crosses. The developed lines with the blast resistance gene exhibited no lesions on leaves and received a representative score of “0–2”. The improved lines with two-gene combinations *(Pi54 + Rf3 & Pi54 + Rf4)* were excluded from our analysis. The blast resistance scoring and pollen fertility status for the improved lines and parents is presented in Table 3. The donor parent, B 95 (B 95-1 × Tetep) with the *Pi54* gene, was found to be resistant to the virulent isolate of the pathogen (IS (KUL)-6) which is more prevalent in South India. At the time of the disease evaluation, each of the improved lines exhibited no symptoms and thus scored as “1”. Roundish-shaped, small gray spots, slightly enlarged (1–2 mm in diameter) with a brown margin, present on the lower leaves were scored as “2”. High pollen fertility was observed in all the improved lines (Appendix A)

### 3.2. Agronomic Traits in Improved Restorer Lines

The twenty-two three-positive BC_3_F_3_ plants, along with the parents of both crosses, were screened for agronomic performance parameters. The three-gene pyramided lines 5-3-7-6 and 5-3-8-10, although phenotypically similar to recurrent parents, performed better for most of the agronomic traits studied (Table 4, Appendix A). Interestingly, some of the improved lines had earlier maturity (8 days) compared to CB 87 R and CB 174 R. Five out of 22 improved lines possessed increased productive tillers, number of filled grains, and hundred grain weight as compared to recurrent parents. None of the improved lines displayed significant variation for plant height, panicle length, and grain yield compared to the unimproved restorer lines.

### 3.3. Cluster Analysis

Agglomerative clustering with Euclidean values divided the improved restorer lines and parents into four clusters (cophenetic correlation coefficient = 0.716). Cluster one consisted of three parents and two improved lines, while cluster two had the donor parent alone. Cluster three had six improved lines and cluster four had fourteen improved lines (Appendix A). The highest yielding improved lines were placed in cluster one and found very similar to both recurrent parents.

## 4. Discussion

The present study was carried out with the objective to introgress a blast resistant gene, *Pi54,* into the background of popular restorer lines, CB 87 R and CB 174 R, through marker-assisted back-cross breeding coupled with screening for improved agronomic traits. Improvement of these lines for blast disease resistance facilitates the development of new generation of hybrids with durable resistance against blast disease in addition to bacterial blight disease. Since the 1960s, more than 100 resistance genes or loci for blast disease have been identified [28,29]. Among them, the majority of the genes have been induced by pathogen infection, while a few of the genes express constitutively. In addition, the expression of many R genes induce resistance against leaf blast at the seedling stage, while only a few R genes confer resistance to panicle blast [30,31,32,33]. Such complex interactions of R genes are an important challenge in blast resistance breeding to achieve broad-spectrum and durable resistance. Earlier reports on pyramiding of *Pi* genes against blast disease in different background combinations (*indica, japonica*) indicated the expression of varying levels of resistance. This means that different gene combinations produce different interaction effects, and some show no resistance comprising the simple accumulation of the resistance spectra of the target R genes [34,35,36,37]. However, the blast-resistant gene, *Pi54,* which was identified in a highly resistant genotype, Tetep, was confirmed to have broad-spectrum resistance against predominant races found in India [2,13]. The *Pi54*, from the *Piz* locus, also shows broad-spectrum resistance to both leaf and panicle blasts [12]. Wu et al. [37] also reported that pyramiding of *Pi54* along with *Pi1, Pi33,* and *Pigm* exhibited high level of resistance to leaf and panicle blasts. Similarly, it was also proved that the R genes of the alleles from the *Piz* locus exhibit excellent resistance after combining them with certain independently distributed R genes, such as *Pi56* and *Pish* [38]. To date, many successful marker-assisted introgression using the *Pi54* gene have been made in the various backgrounds of rice varieties and hybrids all over the world [2,36,39,40,41]. Keeping in view with this, the present study aimed to introgress the *Pi54* gene from Tetep into stable restorer lines CB 87 R and CB 174 R, but susceptible to blast disease through MABB. To the best of our knowledge, ours is the third such report wherein functional marker *Pi54* MAS has been utilized for foreground selection. Earlier, Vijay Kumar et al. [1] successfully introgressed the *Pi54* gene from a highly blast-resistant genotype, i.e., Tetep, into an elite rice cultivar Samba Mahsuri (BPT 5204), a high-yielding rice variety with good cooking quality, but susceptible to blast disease, through MABB. Subsequently, a Samba Mahsuri variety, JGL1798, was introgressed with the *Pi54* gene to resist against blast disease along with BB resistance through functional marker *Pi54* MAS [42]. Therefore, pyramiding the alleles of *Pi54* along with BB-harboring stable restorer lines would greatly accelerate the hybrid rice program for durable resistance against important biotic stresses of rice.

For foreground selection, the functional marker, *Pi54* MAS specific to the *Pi54* gene, was used to identify the plants carrying heterozygous alleles for the targeted resistant gene at each back-cross generation. The *Pi54* MAS is an InDel targeting a 144-bp insertion/deletion polymorphism between the resistant and susceptible allelic sequences [21]. In addition, its co-dominant nature perfectly co-segregates the blast resistance gene in a 1:2:1 ratio in the segregating mapping population. Earlier, studies also proved that the *Pi54* MAS has an accurate genotyping ability (>98%) and predicts the allelic status efficiently in many rice cultivars, and it is routinely used in the many MABB programs [12,21,43] globally. For background analysis, 156 SSR markers distributed across the rice genome were employed to analyze the recovery of recurrent parent genome (RPG) in both improved lines. The stringent phenotypic and background selection from BC_1_, BC_2_, and BC_3_ populations recovered 90.3 to 91.1% recurrent parent genomes at BC_3_F_3_ generation of both populations. While improving the restorer line for blast resistance, it is absolutely necessary that the back-cross-derived lines should also possess the fertility-restorer genes, *Rf3* and *Rf4*. Closely linked gene-specific markers, DRRM-*RF3*-10, DRCG*-RF4-*8, and RM6100, were used for the identification of fertility restoration genes *Rf3* and *Rf4* along with target gene (*Pi54*) in the segregating population.

In this study, we used a higher number of parental polymorphic markers with a better coverage per chromosome for genetic background selection. This has certainly resulted in limiting the linkage drag to the regions closer to the target genes [44]. Through stringent phenotypic and background selection at each back-cross generation, we have identified intercrossed lines with increased productive tillers, grain filling, and 100-grain weight than the recurrent parent. This indicates that utilization of the improved blast-resistant restorer lines in the hybridization program would be of great advantage currently and in the future.

Our research group has previously released two rice hybrids, CO RH3 and CO RH4, utilizing the restorers, CB 87 R and CB 174 R [45]. These hybrids played a significant role in improving the rice production from 2010 onwards at both the regional and national level. On the contrary, there were continuous complaints from the farmers about susceptibility of the above hybrids to the blast disease. Earlier, we improved three popular cytoplasmic male sterile lines, CO 2A, CO 23A, and CO 23B, for bacterial blight resistance through functional markers and developed stable hybrids with high yield and BB resistance [19]. Our improved lines, especially 5-3-7-6 and 5-3-8-10 with the *Pi54* gene, and similar phenotypic characters of restorer lines have brought back the full genetic potential and yield of two hybrids, CO RH3 and CO RH4. In addition, the improved restorer lines and already improved cytoplasmic male sterile lines will be useful in the development of new blast and bacterial blight-resistant hybrids in the near future.

In conclusion, MABB breeding techniques are much effective than conventional breeding, which helps in the quicker and maximum recovery of genetic backgrounds along with target genes with minimum linkage drag. We successfully introgressed a single dominant blast resistance gene, *Pi54,* in the susceptible restorer rice lines viz. CB 87 R and CB 174 R, along with the two fertility-restorer genes *Rf3* and *Rf4*. The introgression of *Pi54* remarkably increased the blast resistance level and maintained the yield levels in blast endemic regions. The breakdown of resistance is a natural phenomenon and the only alternative is to introgress the various allelic forms of *Pi54* in a single genetic background. We expect that the improved lines will provide blast resistance at least for a foreseeable period in Southern India. Meanwhile, our *Pi54* allelic pyramided restorer lines will be ready for long-term use in hybrid rice production.

## Figures and Tables

**Figure 1 genes-11-01266-f001:**
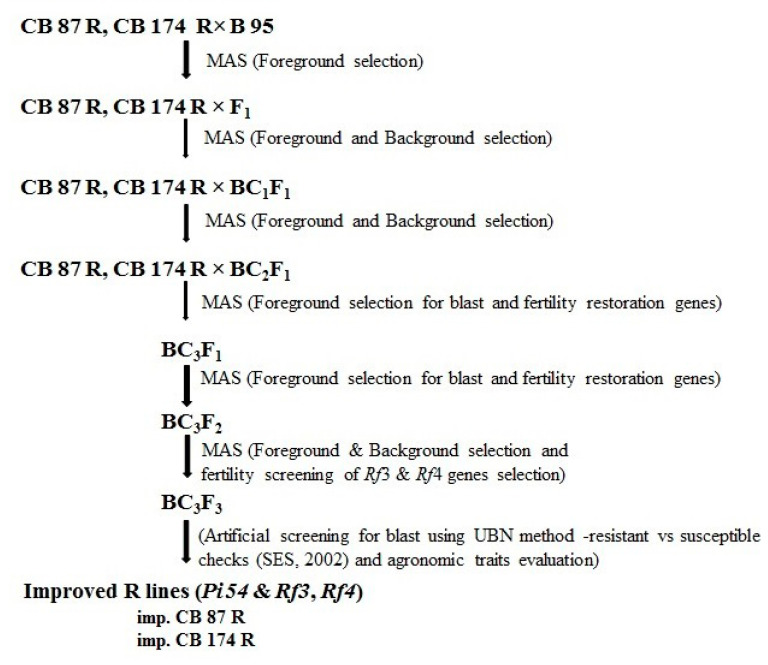
Schematic steps for introgression of blast resistance genes into restorer lines through marker-assisted selection Progenies homozygous for blasts (*Pi*54) and fertility restoration (*Rf3, Rf4*) were evaluated for agronomical traits and agro-morphological traits.

**Figure 2 genes-11-01266-f002:**
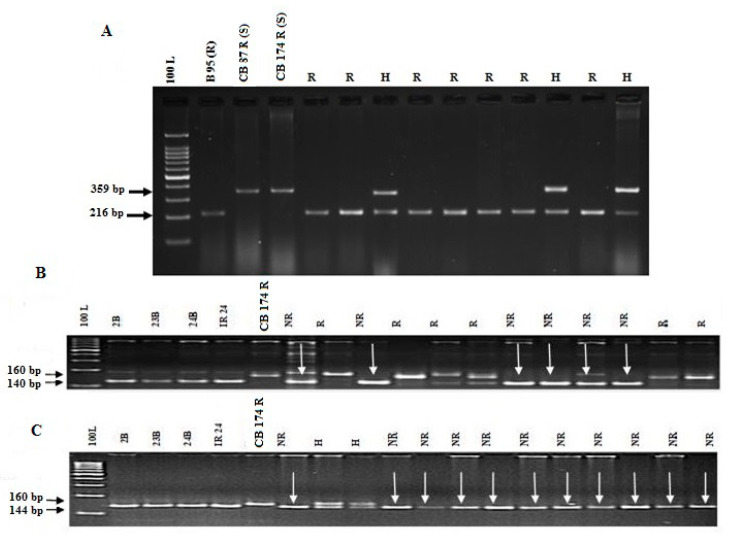
Foreground selection in progenies. PCR amplification of BC_3_F_2_ plants in CB 87 R **×** Tetep cross. (**A**) *Pi*54 MAS, primer for *Pi*54 blast resistance gene. (**B**) DRRM-*RF3*-10, marker for *Rf*3 gene. (**C**) RM 6100 marker for *Rf*4 gene. 100 L, 100 bp ladder; R, resistant; H, heterozygote; S, susceptible; NR, non- restorer; R, restorer. B 95 is the blast resistance line; CB 87 R, CB 174 R are susceptible lines; 2B, 23B, 24 B, IR 24 are non-restorer genotypes; CB 87 R, CB 174 R are restorer genotypes.

**Figure 3 genes-11-01266-f003:**
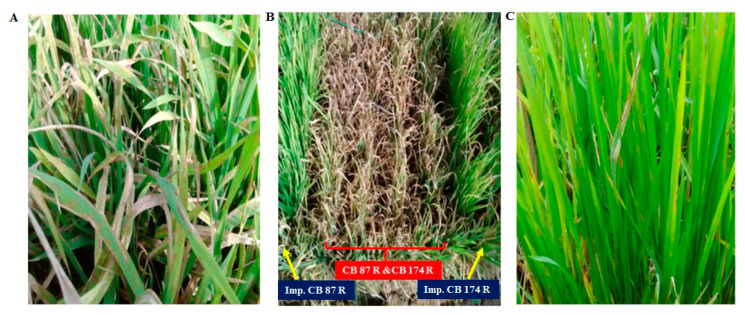
Phenotypic screening for blast resistance. *Magnaporthe oryzae-*infected plants (post-inoculation view). (**A**) Negative check, IR 24. (**B**) Unimproved and improved restorer lines of CB 87 R and CB 174 R. (**C**) Positive check, Tetep.

**Table 1 genes-11-01266-t001:** Details of markers used for blast resistance and fertility restoration.

S.No	Marker Types	Primer	Primer Sequences 5′→3′	Annealing Temperature (°C)	Amplified Product Size (bp)	Base Pair/cM	Chromosomal Location	Citation
**Blast Marker**								
1	Functional	*Pi54* MAS	F- CAATCTCCAAAGTTTTCAGG R-GCTTCAATCACTGCTAGACC	56 °C	359	-	11	[20]
***Rf*_3_ Locus**								
2	SSR	DRRM-*RF3*-10	TCTGTGCATTGCCTGAACAT TCGTATGGAACGATGTGATGA	56 °C	140	4982046	1	[21]
***Rf*_4_ Locus**								
3	SSR	DRCG-*RF4*-8	F-TGGGATCATGAAAGCCATAC R-GCTTTATAGGCGCCGATTTT	57 °C	845	18211995	10	[21]
4	SSR	RM 6100	F- TCCTCTACCAGTACCGCACC R- GCTGGATCACAGATCATTGC	58 °C	160	1.2 cM	10	[22]

**Table 2 genes-11-01266-t002:** Segregating ratio of the marker genotypes in BC_3_F_2_ generation for *Pi*54 blast resistance gene.

S.No	CB 87 R × B 95						CB 174 R × B 95				
	Markers	Observed Frequency					Observed Frequency				
		RR	Rr	rr	Total	χ^2^ (1:2:1)	RR	Rr	rr	Total	χ^2^ (1:2:1)
1	*Pi54* MAS	41	106	63	210	1.504	46	127	87	260	1.487

RR, rr, Homozygotes; Rr, Heterozygote.

**Table 3 genes-11-01266-t003:** Disease screening and fertility status of three-gene positive BC_3_F_3_ improved lines for resistance to blast disease.

S. No	Pyramided Lines	Allelic Status (*Pi54, Rf3, Rf4*)	Resistance Genes Genotyped by Linked Marker	Reaction Against to Blast		Pollen Fertility Status (%)
				* Disease Scoring for Rice Blast (0–9 scale)	R/S	
**CB 87 R × B 95—Improved CB 87 R**						
1	5-3-7-1	*Pi54 + Rf3 + Rf4*	++	1	R	92.1
2	5-3-7-3	*Pi54 + Rf3 + Rf4*	++	1	R	92.8
3	5-3-7-4	*Pi54 + Rf3 + Rf4*	++	0	R	92.3
4	5-3-7-5	*Pi54 + Rf3 + Rf4*	++	2	R	82.6
5	5-3-7-6	*Pi54 + Rf3 + Rf4*	++	1	R	91.2
6	5-3-7-8	*Pi54 + Rf3 + Rf4*	++	1	R	82.3
7	5-3-7-9	*Pi54 + Rf3 + Rf4*	++	2	R	80.1
8	5-3-7-10	*Pi54 + Rf3 + Rf4*	++	2	R	93.8
9	5-3-7-12	*Pi54 + Rf3 + Rf4*	++	1	R	91.7
**CB 174 R × B 95—Improved CB 174 R**						
10	5-3-8-1	*Pi54 + Rf3 + Rf4*	++	1	R	81.1
11	5-3-8-2	*Pi54 + Rf3 + Rf4*	++	2	R	90.8
12	5-3-8-4	*Pi54 + Rf3 + Rf4*	++	0	R	80.4
13	5-3-8-6	*Pi54 + Rf3 + Rf4*	++	1	R	81.7
14	5-3-8-7	*Pi54 + Rf3 + Rf4*	++	2	R	91.6
15	5-3-8-8	*Pi54 + Rf3 + Rf4*	++	2	R	83.9
16	5-3-8-10	*Pi54 + Rf3 + Rf4*	++	2	R	93.8
17	5-3-8-11	*Pi54 + Rf3 + Rf4*	++	0	R	91.2
18	5-3-8-15	*Pi54 + Rf3 + Rf4*	++	1	R	81.4
19	5-3-8-16	*Pi54 + Rf3 + Rf4*	++	1	R	88.9
20	5-3-8-17	*Pi54 + Rf3 + Rf4*	++	1	R	91.2
21	5-3-8-19	*Pi54 + Rf3 + Rf4*	++	1	R	90.3
22	5-3-8-22	*Pi54 + Rf3 + Rf4*	++	1	R	80.9
23	IR 24 (Negative check)		−−	9	S	
24	CB 87 R (Recurrent parent)	*Rf3 + Rf4*	−−	9	S	95.6
25	CB 174 R (Recurrent parent)	*Rf3 + Rf4*	−−	9	S	92.9
26	B 95 (Donor parent)	*Pi54*	++	1	R	
27	Tetep (positive check)	*Pi54*	++	1	R	-

* The back-cross-derived lines at BC_3_F_3_ (*Pi54* + *Rf*3 + *Rf*4) three-line pyramided lines were screened with a blast isolate under controlled conditions using the UBN method. “+” and “−” indicates positive and negative alleles.

**Table 4 genes-11-01266-t004:** Agronomic characters of improved restorer lines.

S.No	Plant No.	Days to Flowering	Plant Height (cm)	No. Productive Tillers	Panicle Length (cm)	No. Filled Grains	100 Grain Weight (g)	Grain Yield (g)
1	5-3-7-1	73.9 ^1^	78.2	13.5	22.7	124.9	2.17 ^1^	24.17
2	5-3-7-3	72.3 ^1^	73.5	12.8	21.6	116.7	1.83	22.87
3	5-3-7-4	73.2 ^1^	76.8	13.7	20.8	118.5	2.07 ^1^	19.43
4	5-3-7-5	71.5 ^1^	84.9	11.5	23.3	138.3 ^1^	2.17 ^1^	22.30
5	5-3-7-6	77.1	84.4	15.8 ^1^	23.1	123.0	1.77	27.07
6	5-3-7-8	71.8 ^1^	78.1	14.4	20.8	109.8	1.73	21.43
7	5-3-7-9	68.6 ^1^	68.6	15.2 ^1^	18.7	128.6	2.13 ^1^	18.73
8	5-3-7-10	71.7 ^1^	77.3	13.1	22.0	128.6	2.30 ^1^	21.23
9	5-3-7-12	75.6	85.5	14.9 ^1^	21.2	123.2	2.03 ^1^	20.17
10	5-3-8-1	73.2 ^2^	75.5	13.4	22.7	110.5	2.17	22.30
11	5-3-8-2	69.1 ^2^	89.1	13.8	18.6	125.6 ^2^	2.27	19.07
12	5-3-8-4	74.7 ^2^	74.7	14.7 ^2^	15.2	115.0	2.13	20.87
13	5-3-8-6	78.7	96.3	13.3	18.4	117.7	2.23	18.47
14	5-3-8-7	71.6 ^2^	77.0	13.3	22.2	123.6 ^2^	1.83	21.77
15	5-3-8-8	75.2 ^2^	75.4	13.6	18.8	116.1	2.07	19.17
16	5-3-8-10	77.5	95.8	16.0 ^2^	24.1	126.8 ^2^	2.40	26.40
17	5-3-8-11	72.7 ^2^	85.8	13.0	17.3	125.6 ^2^	2.13	19.60
18	5-3-8-15	72.2^2^	84.6	12.2	19.0	116.9	2.03	20.13
19	5-3-8-16	71.8 ^2^	81.8	11.3	21.6	117.5	1.97	21.30
20	5-3-8-17	76.2	83.9	12.3	16.1	112.7	1.73	19.63
21	5-3-8-19	71.2 ^2^	76.0	12.4	21.2	120.9	1.97	21.73
22	5-3-8-22	73.3 ^2^	76.7	13.2	22.1	109.3	2.17	20.83
23	IR 24	83.3	86.0	13.3	24.0	119.3	2.33	29.17
24	CB 87 R	76.8	86.8	13.2	22.9	127.5	1.83	27.90
25	CB174 R	77.2	95.5	12.6	23.4	115.5	2.50	26.03
26	B 95	77.4	114.3	12.4	22.9	98.3	2.13	25.60
	Mean ± 2SE	73.11	85.78	14.35	21.97	124.75	3.40	23.46
	CV%	1.16	2.68	6.16	4.42	3.73	3.60	4.89
	LSD	1.4	3.7	1.4	1.5	7.3	0.58	1.78

^1^ Increased over CB 87 R, ^2^ increased over CB 174 R; IR 24 susceptible check, CB 87 R, CB 174 R recurrent parents, B 95 donor parent; CV, coefficient of variation; LSD, critical difference at 5% probability level; values are mean of three replications.

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
