# Peer review of "Improvement of Stable Restorer Lines for Blast Resistance through Functional Marker in Rice (Oryza sativa L.)"

_genes, 2020, doi:10.3390/genes11111266_

Round 1
Reviewer 1 Report
The manuscript entitled “Improvement of stable restorer lines for blast resistance through functional marker in rice (Oryza sativa L.) has important and interesting research work. There are grammatical and other English errors in the manuscript. Besides, the research result should be improved with statistical analysis. Thus, I recommend the manuscript to be accepted after a major revision.
- The manuscript needs to be improved for write up. Professional English editing is needed to improve the manuscript.
- Few mistakes are shown in the manuscript and a scan copy is attached herewith.
- Material methods section need to be improved with detail description of where and how the experiments were done.
- For blast and agronomic data, perform statistical analysis and compare the results.
- Provide the ANOVA.
- It was not mentioned about the experimental location for agronomic data. If the experiment was done in the greenhouse, how reliable the result will be? Provide enough discussion in this regard.

Author Response
Sir,
Thanks for the detailed review. As suggested the manuscript was revised thoroughly and improved for kind consideration.
Response to Reviewer comments
Reviewer 1:
Comment 1: The manuscript needs to be improved for write up. Professional English editing is needed to improve the manuscript.
Answer: The entire manuscript has been improved for English language and other typographical errors.
Comment 2: Few mistakes are shown in the manuscript and a scan copy is attached herewith.
Answer: The suggested corrections was carried out in track-change mode. For instance
- The recurrent parent names, CB 87 R and CB 174 R has been now given uniformly throughout the manuscript
- The Figure 1 mistakenly uploaded was now corrected and re-included
- In Table 1, the RF-3 gene specific marker has been italicized and as suggested one more column on base pair or distance has been included
- The Figure 2 was revised for it correct location of base pair (band size) over the gel picture. The corrected figure 2 was included
- Line number 152: The recovery percentage of the BC1F1 was mistakenly typed as 83.6 and 84.9 % although we got 73.6 and 74.9 %. Now the correction was carried out. Please see the line number 152 of the manuscript.
- Line number 182: The naming of the selected improved restorer lines has been now simplified as suggested. For your kind information, the naming pattern given for 5-3-7-1 means that, 5th F1, 3rd back cross, 7th plant in F2 and 1st- first homozygous line.
Comment 3: Material methods section need to be improved with detail description of where and how the experiments were done.
Answer: The materials and method section was updated with experimental design, replication and statistical analysis carried out.
Comment 4: For blast and agronomic data, perform statistical analysis and compare the results.
Answer: The statistical analysis for agronomic traits along with blast response of the improved lines were detailed through newly inclusion of ANOVA table (Supplementary Table S3) and compared through cluster analysis (Supplementary Figure S2) and bar chart diagram(Supplementary Figure S3).
Comment 5: Provide the ANOVA.
Answer: As suggested, the ANOVA table was included
Comment 6: It was not mentioned about the experimental location for agronomic data. If the experiment was done in the greenhouse, how reliable the result will be? Provide enough discussion in this regard.
Answer: The experiment was conducted in the farm field at Paddy Breeding Station, Tamil Nadu Agricultural University, Coimbatore, India. It was not conducted in the green house. The ANOVA table (Supplementary Table S3) and cluster diagram (Supplementary Figure S2) and bar chart (Supplementary Figure S3) were provided in the supplementary file.
Reviewer 2 Report
The authors describe the introgression of a dominant gene Pi54, which is a known broad-spectrum disease resistance gene against blast, into two elite restorer lines. They have used the marker assisted backcross breeding approach using a gene-specific functional marker, at each of the backcross generations. The authors also used large number of SSRs covering all the twelve chromosomes to recover the maximum recurrent parent genome. At the end of the process they identified the Rf3 and Rf4 target genes, and confirmed blast resistance under stringent phenotyping, along with high yield. The procedure described in the article demonstrates the potential of MAS using functional and SSRs markers.
MAS is a molecular breeding tool that can save laborious works and a precious time. In this study, the authors could have saved more work and time if they would use a high throughput SNP genotyping platform (SNPs array) available for rice, for the background test to achieve maximum recurrent parent genome. The authors should describe the current genomic technology currently available for rice and explain why they preferred to use SSRs (maybe timing…?). Nevertheless, since the authors have successfully applied the MAS approach, I recommend to consider the paper for publication with minor corrections.
The authors provide the SSR list, but they do not provide the source of these SSRs. If these are published SSRs they should provide the references. If they are newly designed the primers, they should provide a list of all sequences of primers.
Author Response
Sir,
Thanks for the positive and detailed comments. It helps to revise the manuscript thoroughly and further improvement. The revised MS may kindly be considered.
Comment 1: The authors should describe the current genomic technology currently available for rice and explain why they preferred to use SSRs (maybe timing…?).
Answer: The SSR makers distributed throughout the 12 chromosomes were used in the current work because of cost, easiness and time. We are now equipping our laboratory with advance equipment like Fluidigm for SNP analysis. Future work will be concentrated on this area.
Comment 2: The authors provide the SSR list, but they do not provide the source of these SSRs. If these are published SSRs they should provide the references. If they are newly designed the primers, they should provide a list of all sequences of primers.
Answer: As suggested, the details of SSR markers source (Gramene URL) was given in Supplementary Table S1. Please see the revised supplementary file
Round 2
Reviewer 1 Report
Thanks for all those necessary changes. I have no further suggestion thus cacn be accepted for publication in its current form.